# Epidemiology of *PAX6* Gene Pathogenic Variants and Expected Prevalence of *PAX6*-Associated Congenital Aniridia across the Russian Federation: A Nationwide Study

**DOI:** 10.3390/genes14112041

**Published:** 2023-11-04

**Authors:** Tatyana A. Vasilyeva, Andrey V. Marakhonov, Anna A. Voskresenskaya, Vitaly V. Kadyshev, Natella V. Sukhanova, Marina E. Minzhenkova, Nadezhda V. Shilova, Alla A. Latyshova, Evgeny K. Ginter, Sergey I. Kutsev, Rena A. Zinchenko

**Affiliations:** 1Research Centre for Medical Genetics, 115522 Moscow, Russia; vasilyeva_debrie@mail.ru (T.A.V.); vvh.kad@gmail.com (V.V.K.); natelasukhanova@gmail.com (N.V.S.); maramin@mail.ru (M.E.M.); nvsh05@mail.ru (N.V.S.); ekginter@mail.ru (E.K.G.); kutsev@mail.ru (S.I.K.); renazinchenko@mail.ru (R.A.Z.); 2Fyodorov Eye Microsurgery Federal State Institution Cheboksary Branch, 428028 Cheboksary, Russia; vsolaris@mail.ru; 3Russian Research Institute of Health, 127254 Moscow, Russia; lat-alla75@mail.ru

**Keywords:** congenital aniridia, *PAX6*, WAGR syndrome, genetic epidemiology, prevalence, unexamined patients

## Abstract

This study investigates the distribution of *PAX6*-associated congenital aniridia (AN) and WAGR syndrome across Russian Federation (RF) districts while characterizing *PAX6* gene variants. We contribute novel *PAX6* pathogenic variants and 11p13 chromosome region rearrangements to international databases based on a cohort of 379 AN patients (295 families, 295 probands) in Russia. We detail 100 newly characterized families (129 patients) recruited from clinical practice and specialized screening studies. Our methodology involves multiplex ligase-dependent probe amplification (MLPA) analysis of the 11p13 chromosome, *PAX6* gene Sanger sequencing, and karyotype analysis. We report novel findings on *PAX6* gene variations, including 67 intragenic *PAX6* variants and 33 chromosome deletions in the 100 newly characterized families. Our expanded sample of 295 AN families with 379 patients reveals a consistent global *PAX6* variant spectrum, including CNVs (copy number variants) of the 11p13 chromosome (31%), complex rearrangements (1.4%), nonsense (25%), frameshift (18%), and splicing variants (15%). No genetic cause of AN is defined in 10 patients. The distribution of patients across the Russian Federation varies, likely due to sample completeness. This study offers the first AN epidemiological data for the RF, providing a comprehensive *PAX6* variants spectrum. Based on earlier assessment of AN prevalence in the RF (1:98,943) we have revealed unexamined patients ranging from 55% to 87%, that emphases the need for increased awareness and comprehensive diagnostics in AN patient care in Russia.

## 1. Introduction

Congenital aniridia (AN, OMIM #106210) is a severe inherited disease that visibly affects the iris, leading to its partial or total absence. This condition also critically impacts various eye structures in both the anterior and posterior segments [1]. The main causative factor for congenital aniridia is pathogenic variants in the *PAX6* gene [2]. These variants include intragenic *PAX6* pathogenic variants and large chromosome rearrangements encompassing the 11p13 region, which harbors the *PAX6* gene or only its downstream regulatory regions [3,4]. Some large chromosome deletions may also involve the neighboring locus of the *WT1* gene, giving rise to WAGR syndrome (OMIM #194070). WAGR syndrome is characterized by the presence of Wilms tumor (W), congenital aniridia (A), genitourinary anomalies (G), and mental retardation (R) with eye abnormalities [5].

Studying the distribution of genetically determined diseases holds immense significance. Despite their rarity, orphan conditions can collectively contribute significantly to the overall disease burden [6,7]. Understanding their prevalence is crucial for optimizing resource allocation, planning medical assistance, and providing social support to affected families. From both an economic and humanitarian perspective, addressing these genetic diseases is of utmost importance [8,9].

The molecular genetic study of AN and WAGR syndrome in Russia began a decade ago in response to the request of geneticists, ophthalmologists, and AN patients’ families. Initially, our expectations were modest, but over the years, we have successfully achieved extensive coverage of Russian families with AN. Our study encompasses various federal districts of the Russian Federation, including the Central, Northwestern, Volga, Ural, Southern, North Caucasian, Siberian, and Far Eastern districts.

The primary objectives of our study were to estimate the prevalence of *PAX6*-associated AN cases across the entire country, identify potential regional variations, and explore the underlying reasons for such differences. To achieve these goals, we have focused on establishing genetically confirmed cases of AN, as the defined genetic cause of AN serves as the most accurate diagnostic criterion.

## 2. Materials and Methods

### 2.1. Samples

#### 2.1.1. The Study Sample

We conducted an analysis of PAX6 gene variant frequencies and the prevalence of congenital aniridia (AN) and AN-related cases within the population of the Russian Federation (RF). Our study sample consisted of 379 patients from 295 families who have received clinical diagnoses and genetic confirmation since the inception of our study in 2011. This cohort of patients comprised two distinct groups: (i) 129 newly included patients from 100 new families who were included in our ongoing study; (ii) 250 previously studied patients from 195 families who had previously been described in the earlier phases of our research [10]. The patients for our study are recruited through various routes, including genetic counseling sessions at the Research Center for Medical Genetics in Moscow (RCMG). Additionally, our team of geneticists from the Laboratory of Genetic Epidemiology and the Scientific Counseling Department at RCMG conduct expeditions to the Northern Caucasus district to identify and include patients in the study.

The Cheboksary Branch of the S. Fyodorov Eye Microsurgery Federal State Institution has made a significant contribution to our patient recruitment efforts, with approximately one-third of all patients initially diagnosed with congenital aniridia originating from this source.

In order to broaden our outreach and ensure comprehensive coverage, RCMG has initiated a dedicated scientific program focused on the study of congenital aniridia. This program encourages ophthalmologists encountering patients with congenital aniridia to refer them to our center for molecular diagnostics. Our primary goal with this program is to raise awareness of molecular genetic diagnostic options among both medical professionals and affected families.

To safeguard the rights and privacy of participants, each patient or their legal representative is required to sign an informed consent form before participating in the study. Ethical approval for the study has been granted by the Ethics Committee of the Research Centre for Medical Genetics (Protocol #5, dated 10 December 2010).

By utilizing these diverse recruitment channels and adhering to ethical guidelines, we are steadfast in our commitment to conducting a comprehensive and collaborative study on congenital aniridia. Our aim is to advance our understanding of this condition and enhance diagnostic and support services for affected individuals and their families.

In the total sample, the ratio of single cases to familial cases is 217:162, and the ratio of male to female patients is approximately 1:1.15. The average age of patients in the cohort is 16.9 ± 16.8 years old.

#### 2.1.2. Epidemiology Study Data

The epidemiology study sample, consisting of 3,195,054 residents of the Russian Federation (RF), was obtained from a prior genetic epidemiology study conducted by our laboratory in 14 regions of the European part of the Russian Federation. These regions include Kirov, Kostroma, Tver, Bryansk, Rostov oblasts, Krasnodarsky kray, the Chuvash and Udmurt Republic, the Republics of Tatarstan, Bashkortostan, Northern Ossetia–Alania, Adygea, and Karachay-Cherkessia. This genetic epidemiology study involved the surveying of 3,195,054 individuals for hereditary diseases using a comprehensive protocol specifically designed for complex genetic-epidemiology studies. A detailed description of this protocol can be found elsewhere [8].

Briefly, the genetic epidemiology survey was conducted in three consecutive steps. At the first step, local medical services complete a questionnaire with the patients’ disease anamnesis information. The “Multiple registration” method was used [11], where information could be obtained from several sources. At the second step, the patients were examined by clinical geneticists from the RCMG. At the third step, clinical investigations were performed by different specialists. In the previous genetic-epidemiology study, the expected prevalence of AN (Aniridia) cases in the Russian Federation was determined to be 1 in 98,943 (1.011 per 100,000 individuals) [unpublished data]. This valuable data serves as a basis for comparison and as a reference point for the prevalence assessment of AN in our primary study sample. It is worth noting that the *PAX6* cases identified in this group are also included in the study sample (Section 2.1.1).

#### 2.1.3. Official Statistical Population Data

We also incorporate data from medical care officials, specifically the annual summary report for 2021 from the Republic of Chuvashia Ministry of Public Health, which we obtained from its official source [https://oftalmolog.med.cap.ru/patient-rights/otcheti, accessed on 31 August 2023]. According to this report, in 2021, out of 28,968 individuals seeking eye medical care for the first time in the Chuvashia Republic, 8751 were referred for diagnosis to the Cheboksary Branch of the S. Fyodorov Eye Microsurgery Federal State Institution. This implies that the doctors at the collaborative Ophthalmological Center can only observe a third of the total number of expected patients. We have applied a correction coefficient of 0.3, representing this portion, to calculate the potential number of all expected WAGR patients in the studied districts, taking into account patients who are not within our immediate observation.

### 2.2. DNA Isolation

DNA was extracted from blood samples obtained from 129 aniridia patients, comprising 100 families. This extraction was performed using the Wizard Genomic Purification Kit from Promega, Madison, WI, USA, following the manufacturer’s recommended protocols.

Subsequently, all biological samples were securely stored in the Moscow Branch of the Biobank “All-Russian Collection of Biological Samples of Hereditary Diseases”.

### 2.3. Molecular Genetic Diagnostics

Multiplex ligase-dependent probe amplification analysis (MLPA) of gene copy number variation in the 11p13 chromosome region using a MRC Holland probe kit P219 according to the manufacturer’s recommendations followed by Sanger sequencing of the PAX6 gene exons is performed, as described earlier [10].

Karyotype analyses are performed in 4 cases without findings by MLPA and Sanger sequencing.

### 2.4. Statistical Analysis

In our statistical analysis, we employed the Fisher exact test as the statistical criterion when comparing the relative frequencies and distribution of different types of PAX6 pathogenic variants. Additionally, we utilized the Student *t*-test when comparing the frequencies of aniridia (AN) in different districts. These analyses were conducted using WinPepi v. 11.65 [12].

To calculate the expected number of disease cases (*N**) in each district and in the entire sample, we considered the population size (*P*) of the district (or the cumulative population of the studied districts) and the frequency of one case per 98,943 individuals, which was obtained from a previous genetic epidemiology study. The ratio of observed cases (*N*) to expected cases (*N**) allowed us to determine the proportion of patients under observation, while (1 − *N*/*N**) indicated the deficit of cases and the local sample representativeness.

## 3. Results

### 3.1. Clinical Characteristics of a New Sample Comprising 100 Aniridia Families, Totaling 129 Patients

A total of 129 patients (13.1 ± 13.1 years; 39.5% males) were newly included into the study. Nystagmus was observed in 105/120 (87.5%) patients, iris coloboma in 4/129 (3.1%) patients, hypoplastic iris in 4/129 (3.1%) patients, partial aniridia with iridis remnants in 31/129 (24.0%) patients, and complete aniridia in 90/129 (69.8%) patients. Aniridia-associated keratopathy was in 63/112 (56.3%), cataract in 79/110 (71.8%), glaucoma in 33/106 (31.1%), fovea hypoplasia in 93/96 (96%). That was in accordance with other sample data [1,13,14], though the percentage of glaucoma and aniridia associated keratopathy could be lower, probably, due to the lower average age of the patients in our new sample. Among clinical peculiarities, a considerable portion of syndromic patients with severe clinical picture should be mentioned. Apart from 12 WAGR patients, 7 out of 113 patients with an AN diagnosis had either heavy neurological phenotype, or considerable mental retardation with behavioral disorders, or polydactyly, or craniofacial dysmorphias, or hearing loss, or multiple hemangiomas, or other additional symptoms involving different systems. The cause of the phenotypes in all 7 syndromic cases are intragenic *PAX6* loss-of-function variants. Other systems involvement in *PAX6*-associated AN is widely discussed and even observed in rare AN cases [15,16,17,18,19]. In the whole sample there were 21 AN syndromic cases. The portions of syndromic AN in the new sample (6.2%) and in the whole sample (6.1%) were about the same. Two patients with a diagnosis of essential mesodermal iris dystrophy with ectropion uvea associated with complex chromosome 11p13 rearrangements, and two more with initial diagnosis of optic nerve atrophy associated with PAX6 missense variant, constituted the group of patients with so-called AN-related phenotypes. Earlier, the *PAX6* missense variant was defined in two other patients with initial diagnosis of achromatopsia. Thus, in the whole sample, the group of patients with AN-related phenotypes included six persons with non-classical AN phenotypes. Table 1 displays the percentages of patients exhibiting each essential aniridia phenotype feature within the whole sample of 379 patients.

### 3.2. New Data on PAX6 Gene Variants Have Been Identified in a Sample Comprising 100 Aniridia Families, Totaling 129 Patients

In our study of 100 newly characterized aniridia (AN) families, we identified a total of 67 intragenic *PAX6* variants and 33 chromosome rearrangements affecting the 11p13 region. Among the intragenic *PAX6* variants, we observed 19 insertions or deletions (indels) and 48 single nucleotide substitutions. Within this group of 67 variants 22 were novel, previously unpublished small intragenic pathogenic *PAX6* variants, comprising seven novel substitutions and 15 novel indels (Appendix A). All these variants have been assessed in accordance with the ACMG recommendations [20], and they have been classified as either pathogenic or likely pathogenic.

Among the chromosome rearrangements, we detected 33 large 11p13 deletions ranging from 0.2 to 7500 kb in length, with 12 deletions in the range of 2200 to 7500 kb encompassing the “WAGR-region” [10].

In three new cases, we identified complex rearrangements. One case involved a balanced translocation t(10,11)(p15;p13) and a pericentric inversion inv(11)(p12q12), accompanied by a deletion in the “WAGR region” (hg18:chr11:g.(31285887_35117390)del) of at least 3832 kb in length. This rearrangement led to the loss of several genes, including *DCDC1*, *ELP4*, *PAX6*, *RCN1*, *WT1*, *HIPK3*, *LMO2*, and *CD44*. This patient was diagnosed with WAGR syndrome.

In another case, we found a balanced translocation t(2;11)(q34;p15) in a girl diagnosed with AN, even though her parents were healthy. MLPA analysis did not reveal any loss of genetic material in the 11p13-14 chromosome region, suggesting a potential position effect due to the involvement of the *PAX6* gene region in the translocation to the 2q34 region. The breakpoints were not defined exactly, thus, the translocation could either remove regulatory regions from the coding part of the PAX6 gene or remove a part of the rest of the *PAX6* gene disrupting the unity of that with the effect of the deletion of several *PAX6* exons.

Lastly, we identified one more complex rearrangement in a boy with AN, despite his parents being healthy, although his mother had a history of recurrent miscarriages. This complex chromosome 11 rearrangement involved a pericentric inversion inv(11)(p13q14), a paracentric inversion inv(11)(p14p13), and an 11q deletion (11q23.3_11q25.1del). Again, MLPA analysis did not indicate any loss in the 11p13-14 region, suggesting a position effect due to the involvement of the *PAX6* gene region in the pericentric/paracentric inversions. In a previous study, we also identified another complex rearrangement, an 11p13 deletion coupled with a chromosome 11 pericentric inversion inv(11)(p14p13) in a patient with congenital aniridia (A-36 in Table 2) [21].

In summary, we identified complex chromosome rearrangements in four probands, representing 1.4% of the total sample of 295 cases. These rearrangements encompassed two translocations, specifically t(10,11) and t(2;11), as well as three inversions of chromosome 11. One of these inversions was pericentric and, in another case, there was a combination of pericentric and 11p paracentric inversions, accompanied by an additional interstitial deletion in 11q (see Table 2 for details).

### 3.3. Summarized Data on PAX6 Variants Defined in a Broadened Sample (All in All 295 Aniridia Families with 379 Patients)

In the entire sample of 295 probands, we identified *PAX6* small variants in 197 individuals. Specifically, single nucleotide substitutions were found to be the cause of aniridia in 135 probands, and Indels were identified as the underlying genetic alteration in 62 probands. Moreover, large chromosome deletions and other significant rearrangements were detected in 88 probands. These deletions varied in size, ranging from 0.2 Kb to 7500 Kb. Notably, deletions encompassing the “WAGR-region” were observed in 28 probands (Figure 1).

The underlying genetic cause failed to be identified in 10 AN patients.

In our comparison of the previously obtained *PAX6* variant type spectra with the newly acquired data, we have observed a close match. The proportions of variant types have remained consistent. Subsequently, we conducted an in-depth analysis to examine the distribution of different types of *PAX6* variants, including WAGR region deletions, across RF districts. Statistically significant differences in distribution of variant types between regions also were absent. Notably, the recent update has revealed that the prevalence of complex chromosome rearrangements involving 11p13 has increased to 1.4% in the current dataset.

### 3.4. Expected Number of Patients with Congenital Aniridia across the Russian Federation

Based on previous data (unpublished), AN prevalence was estimated to be 1:98,943 people, which is relatively close to the prevalence of 1:76,335 estimated by Orphanet [22]. The expected prevalence of AN in the entire Russian Federation (RF) population, based on these estimates, would be approximately 1480 patients given the population size of 146,507,718 people.

However, in our current study, we identified only 341 genetically confirmed AN cases, and 10 more remained unconfirmed. If we simply divide the number of cases (N = 351) by the total size of the RF population (146,507,718 people), we obtain an underestimated value of 1 in 417,400. To account for the age range in our cohort (0 to 65 years old) and focus on individuals under 65 years old, we adjusted the population size to 123,066,483 people (approximately 84% of the total population) (https://rosstat.gov.ru, Accessed on 31 August 2023) [23]. This adjusted prevalence, based on the adjusted population size, is approximately 1 in 350,617. However, even with this adjustment, we should theoretically observe *N** = 1244 patients with AN.

To reconcile the observed (*N* = 351) and expected (*N** = 1244) numbers of AN patients, we postulate that the majority of AN patients do not come to the attention of genetic counseling services in Russia. We refer to this phenomenon as “data deficit”. The data deficit results in a ratio of cases out of sight to be approximately 72.6% (1 − *N*/*N**) across the entire RF, while the representativeness is approximately 27.4%. The data deficit also varies across different RF Federal districts, and this information is presented in Table 3. Table 3 allows inference on local sample representativeness based on the assumption that (i) AN prevalence is the same in different parts of the country; (ii) there are regional differences in the representativeness between the districts. The observed prevalence of AN in our study appears to be significantly lower than the expected prevalence, suggesting that a substantial proportion of AN cases may not be captured by genetic counseling services in Russia. This variation in data deficit across different regions highlights potential regional differences in representativeness (Table 3).

Among the studied RF federal districts, the Volga and Central districts are the most comprehensively examined. In these districts, we identified a total of 110 and 85 AN cases, respectively. The remaining districts had fewer cases, ranging from 23 to 40. Notably, the Far Eastern federal district had the lowest representation, with only 9 cases.

The ratio of out-of-sight patients varies across districts. In the Volga district, we were able to identify 45% of the expected patients. However, in the Far Eastern district, our data captured only 13% of the expected cases. In other words, the level of underdiagnosis or underreporting is more pronounced in the Far Eastern district compared to the Volga district.

In the well-studied Volga district, the expected AN prevalence assessment (counting patients-out-of-sight) is 1.62 per 100,000. That value is notably close to the data reported by Orphanet for the global population, which is 1.31 per 100,000 [22]. Though the prevalence assessment is slightly higher than those established in previous epidemiological studies for European countries. For instance, in Norway and Sweden, the prevalence rates are reported as 1:76,000 (1.32 per 100,000) and 1:70,000 (1.43 per 100,000), respectively [20].

## 4. Discussion

We have collected data from patients with *PAX6*-associated aniridia, aniridia-related phenotypes, and WAGR syndrome residing in the Central, Northern Western, Volga, Ural, Southern, North Caucasian, Siberian, and Far Eastern federal districts of the Russian Federation (RF). This cohort represents a total population of 146,507,718 people [24].

All of these families have sought genetic diagnostics at the Research Centre for Medical Genetics (RCMG) over the past decade, from 2011 to 2023. In this publication, we present, for the first time, comprehensive data on the genetic causes of PAX6-associated aniridia in a cohort of 100 probands (129 patients) who were newly referred for molecular genetic diagnostics. Additionally, we include data from an expanded and diverse sample comprising 295 families (379 patients) from the RF. This study was conducted as a single-center study, and the findings provide valuable insights into the genetic underpinnings of these conditions in the Russian population.

A summary of the data on PAX6 variants in the expanded sample, which includes 295 aniridia families with 379 patients follows.

*PAX6* small variants were identified in 197 out of the 295 probands. Among these small variants, single nucleotide substitutions were responsible for aniridia in 135 probands and small indels (insertions or deletions) were found to be the cause of aniridia in 62 probands. Large chromosome deletions or other significant rearrangements in the *PAX6* region were identified as the cause of aniridia in 88 probands. Within the group of chromosome deletions only *PAX6* DDR deletions, not encompassing its coding sequence, were defined in 23 probands. The deletion removes the very important enhancers region, which also led to the aniridia phenotype [25]. Within the total sample, chromosome rearrangements that encompassed the “WAGR-region” were detected in 28 probands. These findings provide valuable insights into the genetic mechanisms underlying aniridia in this diverse and comprehensive sample. In 10 AN families, no genetic cause has been defined.

The spectra of *PAX6* variant types in the previous, novel, and entire cohorts exhibit a close match and resemble the world population spectrum. Notable characteristics of the variant spectrum include a high proportion of large 11p13 chromosome rearrangements, accounting for 31% of cases. This category includes complex chromosome rearrangements in four cases, which represents 1.4% of the total. Other significant variant types in the spectrum include nonsense variants (25%), frame shift variants (18%), and splicing changes (15%). For a detailed representation of the data for the entire cohort, please refer to Figure 1.

The most frequent underlying cause of congenital aniridia in the cohort is aggregated small *PAX6* intragenic variants, including single-base substitutions and indels, which were identified in 197 out of 295 families. These variants encompass diverse types, but a significant majority (176 out of 197) result in the formation of premature termination codons (PTCs) and loss of function.

*PAX6* variants are distributed throughout the coding sequence (158 out of 197 cases), with some located in the 5′ untranslated region (5’UTR) in 11 cases and others within introns in 28 cases. Functional studies of variants situated in noncoding regions have confirmed their pathogenic role.

In the most thoroughly studied Volga district, which is represented by 110 AN cases, we observed probands with all possible types of *PAX6* pathogenic variants and chromosome 11p13 deletions. However, in the other districts, we may have failed to identify some rare or even frequent types of variants. Nevertheless, there is a clear trend indicating the presence of nonsense and frame shift variants, as well as large 11p13 deletions, as significant causes of AN in every district. These findings provide insights into the distribution of different *PAX6* variant types and AN prevalence across various districts of the Russian Federation.

The distribution of chromosome deletions in different districts shows some variations, although statistical confirmation is challenging due to the relatively small and potentially unrepresentative population samples in some areas. The relative frequency of chromosome deletions appears somewhat higher in the Far Eastern, Southern, and Ural districts, with ratios of LD (large deletions) to small intragenic variants being 1.17, 0.81, and 0.76, respectively, compared to the frequency in the total sample (0.45). However, these differences do not reach statistical significance.

It is important to note that there is significant variability in the representativeness of local samples across districts. To estimate the number of AN patients who may not be within our observation, we rely on earlier data. In a previous genetic-epidemiology study involving 3,195,054 inhabitants of 14 regions of the Russian Federation, the expected prevalence of AN was assessed as 1:98,943 (1.01 per 100,000 individuals) [8]. Based on these prevalence estimates, we currently observe 45% of expected patients in the Volga district and only 13% in the Far Eastern district. Many confounding factors may influence discrepancy between expected and collected patients. For example, the quality and availability of medical services or unwillingness of the patients to disclose their diagnosis or to be examined.

Given the previously obtained mean AN prevalence of 1:98,943, estimated values of AN prevalence across the RF districts exhibit significant variations. The prevalence of AN in the child population could be much higher. The prevalence ranges from 1:61,726 (1.62 per 100,000) in the Volga district to 1:202,182 (0.49 per 100,000) in the Far Eastern district. It is important to note that smaller sample sizes result in lower prevalence estimates. The limited data available for the Far Eastern district, where only 13% of potential cases are observed, suggests that the estimation based on this small sample may not be very reliable. Conversely, in the well-studied Volga federal district, the AN prevalence assessment of 1.62 per 100,000 closely aligns with data from Orphanet established for other countries (1.31 per 100,000).

When comparing the expected AN prevalence in the well-studied Volga district based on different sources, we find that according to official data, AN prevalence in the Volga district should be approximately 1:74,595 people (323 cases among 24,094,401 individuals). This estimate closely aligns with the expected prevalence based on your earlier AN frequency assessment, which was calculated as 1:59,140. These two estimates are quite similar and suggest a relatively consistent understanding of AN prevalence in the Volga district.

The number of WAGR syndrome patients in the Russian population is proportionally consistent with the number of AN cases, making up 9.5% (28 out of 295) of all cases with AN. This suggests that while the prevalence may be lower, the relative proportion of WAGR syndrome cases to AN cases remain relatively consistent in the Russian population.

## 5. Conclusions

In conclusion, the screening for *PAX6* gene variants in a large cohort of 379 patients with aniridia (AN) has provided valuable insights into the variant spectrum.

Based on previously assessed AN prevalence, a significant portion, ranging from 55% to 87% of the actual number of AN patients residing in the RF remains unobserved. This deficit could have substantial implications, potentially limiting the opportunities for these patients to receive necessary medical and social assistance. Further efforts in understanding and addressing this deficit are crucial for improving the care and support provided to individuals with AN in the Russian Federation.

## Figures and Tables

**Figure 1 genes-14-02041-f001:**
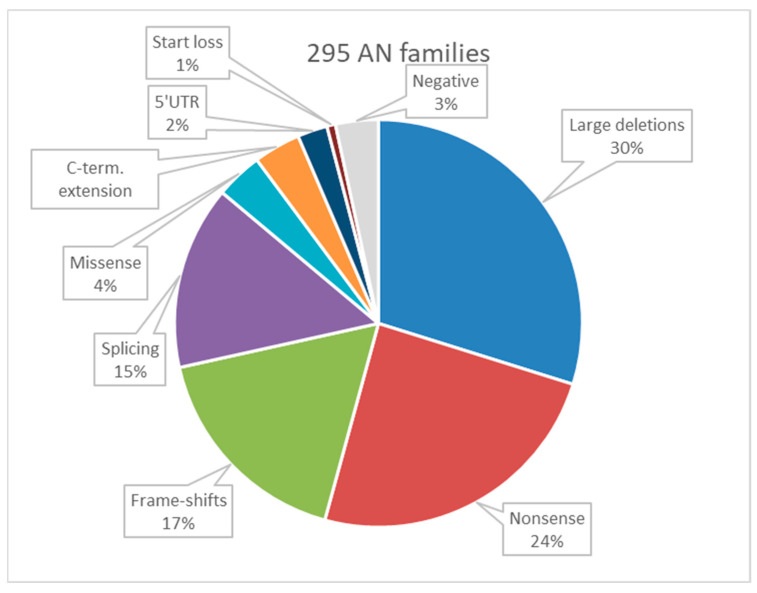
Summarized graphical overview of PAX6 mutation types identified in the expanded sample of 285 aniridia families, comprising a total of 379 patients from the Russian Federation.

**Table 1 genes-14-02041-t001:** Clinical Characteristics of the Sample—Proportions of Patients with Essential Aniridia Phenotype Features.

Aniridia Features	Defined in *N* Patients	Out of the Studied *N* Patients with Data on the Feature	Portion of the Patients with the Feature Defined in the Study	Portion of the Patients with the Feature Defined Earlier
Complete absence of iris	256	318	80.5%	78%
Partial absence of iris/iris hypoplasia/coloboma	62	318	19.5%	22%
Cataract	247	311	79.4%	80%
Foveal hypoplasia	244	264	92.4%	85%
Keratopathy	182	307	59.3%	58%
Nystagmus	235	280	83.9%	78%
Glaucoma	78	318	24.6%	26%

**Table 2 genes-14-02041-t002:** 11p13 complex rearrangements defined in the cohort.

Patient	Karyotype	Visible Event	Additional Events	Deletion in 11p13 (hg18)
t-384.02	46,XY	t(10,11)(p15;p13)	inv(11)(p12q12)	g.(31285887_35117390)del
t-509.04	46,XX	t(2;11)(q34;p15)	nd ^1^	nd ^1^
t-92.03	46,XY	inv(11)(p13q14)	inv(11)(p14p13) + 11q23.3_11q25.1del	nd ^1^
A-36	46,XY	inv(11)(p14p13)	nd ^1^	g.(31628232_32296427)del

^1^ nd—not defined.

**Table 3 genes-14-02041-t003:** Demographic Data on Population of Studied RF Federal Districts and Numbers of AN Cases (Excluding WAGR Syndrome Cases).

District	Age-Adjusted Population Size (under 65 y.o.)	Observed Number of AN Cases (*N*)	Expected Number of AN Cases (Based on Established Prevalence, *N**)	Ratio of Out-of-Sight Cases (1 − *N*/*N**)
Northwestern	11 648 915	24	118	0.80
Central	33 793 164	85	342	0.75
Volga	24 094 401	110	244	0.55
Southern	14 043 291	33	142	0.77
North Caucasian	8 571 229	27	87	0.69
Ural	10 294 912	40	104	0.62
Siberian	13 982 372	23	141	0.84
Far Eastern	6 638 194	9	67	0.87
**Total**	**123 066 483**	**351**	**1244**	**0.72**

## Data Availability

The datasets used and/or analyzed during the current study are available from the corresponding author upon reasonable request.

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
