# Peer review of "Epidemiology of *PAX6* Gene Pathogenic Variants and Expected Prevalence of *PAX6*-Associated Congenital Aniridia across the Russian Federation: A Nationwide Study"

_genes, 2023, doi:10.3390/genes14112041_

Round 1
Reviewer 1 Report
Comments and Suggestions for Authors
General
· Pathogenic variant is the preferred term instead of mutation.
· I have serious concerns about the methods applied to determine prevalence estimates
o 1) Overall AN prevalence: without querying every individual in a population, you should not expect the number of identified cases to give you an estimate of prevalence- there are too many confounding factors when you rely on referrals for collecting individuals
o 2) Applying the 0.3 correction coefficient to WAGR syndrome is also problematic
o 3) Section 2.5 needs to be removed.
· The overview of cases and variants is still useful. I suggest that the paper be modified to note that the number of cases is lower than would be expected based on the published prevalence, suggesting that additional cases are not being referred. Including the epidemiology-based prevalence data (from the second sample) is okay, but more details need to be included about the methods
· Ideally, phenotypic information should be included in more detail. The authors note both AN and ‘AN-related’ in the methods, but do not provide details and the frequency of AN vs ‘AN-related’. Ideally, specific diagnosis information should be added the supplemental table and a general breakdown of the numbers in each group should be included in the results. A review of genotype/phenotype correlations/trends would be more valuable.
Introduction
· In discussing the mechanisms of PAX6 disruption, deletion of the downstream regulatory region (within ELP4) should be mentioned.
Results
· The authors note that they only include PAX6 positive cases. Can any information be provided about the frequency of PAX6 negative cases?
· Line 69: ‘Additionally, we uncovered 22 previously unpublished’ is misleading as seems to suggest 22 cases in addition to the 100 newly characterized described in the paragraph below, but I believe the authors mean to say that within this cohort, 22 variants are novel. Suggest merging this with the paragraph below and using something like ‘within this group of intragenic variants, 22 were novel.’
· Regarding the balanced translocations, discussion of the regulatory region should be included. For example, if the regulatory region is on one side of the breakpoint and PAX6 on the other, this would be a likely mechanism. Alternatively, if the breakpoints fall within the gene, it would result in partial gene deletion. If the breakpoints are not able to be mapped precisely, discussion of the possible mechanisms would be sufficient.
· According to the supplemental table, one of the deletions affects the regulatory region only – this should be highlighted.
· Section 2.3 and Table 2 could probably be omitted. You could consider a statement that the distribution of variant types did not differ among regions, but it would be unexpected if they did.
· Section 2.4- caution against using ‘existing patients’ to describe predicted prevalence. ‘expected patients’ is more accurate
· Section 2.5: I do not feel that these calculations are statistically valid. If comprehensive evaluation was not completed, prevalence cannot be estimated. It is accurate to compare collected cases to predicted based on published prevalence, but the authors cannot use the number of cases collected by referral to calculate prevalence values. Suggest removing this section and focusing on the number of expected missed cases in Section 2.4.
· Instead of comparing variant distribution between regions, it would be more informative to compare the distribution of variant types to other published PAX6 cohorts.
· Similarly, section 2.6 should be modified/removed (see additional notes in methods)
Discussion
· Lines 234-243 simply restate results and should be removed. The following paragraph of the overall distribution is more relevant.
Materials and Methods:
· Per Genes format, this section should be moved between the Introduction and Results, not at the end.
· Line 324-326 is unclear. The authors state that these 100 patients are both new and part of the initially included samples.
· Lines 356-361 seem to belong in results rather than methods
· 4.1.2: second sample. The second paragraph in the section belongs in the results. The first paragraph needs more information on how the prevalence was calculated. Additionally, it is not clear whether the PAX6 cases identified in this group are also included in the group above- this should be specified. I suggest removing the title of ‘second sample’ and replacing with ‘epidemiology study data’ or something similar…
· 4.1.3: first of all, it is not clear why the 0.3 correction was only applied to WAGR and not to AN as a whole. Secondly, I do not believe this is statistically sound- there could be many reasons why patients are not referred for diagnosis, including that less complicated cases are less likely to be referred, which would mean that AN/WAGR cases would be more likely to be referred.
Table 5 (will need to be renumbered)
· Fovel hypoplasia has a typo- currently states 2244
· Would be nice to add glaucoma to this table
· This table belongs in results, not methods.
Comments on the Quality of English Language
The paper is generally well written but has some areas with incorrect word choice/article use. For example:
· Line 36: ‘This condition also cryptically impacts various eye structures in both the anterior and posterior segments’ cryptically is not the correct word choice here- perhaps the authors meant ‘critically’?
· Line 41, 42: ‘the WAGR syndrome’ should just be ‘WAGR syndrome’.
· Line 398-400: implied is not the correct word choice: ‘followed by Sanger sequencing of the PAX6 gene exons is implied, as described earlier’ and ‘Karyotype analysis are implied in 4 cases
Author Response
General
Q1.1· Pathogenic variant is the preferred term instead of mutation.
A1.1 Corrected.
Q1.2· I have serious concerns about the methods applied to determine prevalence estimates
A1.2 Right you are. We attempted to draw attention to the fact that we only estimate expected prevalence and do not determine or define real prevalence. The methods of getting the coefficients, which could predict expected prevalence, were based on real statistical data on the patients referring for the diagnoses. The aim of the project was to predict the number of unobserved patients (so called “out-of-sight cases”) in the country and at least minimal expected prevalence.
Q1.3 1) Overall AN prevalence: without querying every individual in a population, you should not expect the number of identified cases to give you an estimate of prevalence- there are too many confounding factors when you rely on referrals for collecting individuals
A1.3 Overall AN prevalence in Russia was obtained in previous genetic epidemiology studies when every individual in a population was under observation. The methodology as well as detailed results could be found in referred publications [Zinchenko, et al., 2020; Zinchenko, et al., 2021; Kadyshev, et al., 2022]. This overall AN prevalence estimation as 1 per 98,943 was used both to assess the expected number of patients and so called “out-of-sight cases”. We totally agree that many confounding factors could play a role in the discrepancies between observed and expected number of patients. They were suggested to be able to decrease referrals 72% that also could considerably influence expected prevalence. We added these factors in the discussion section under limitation of the study paragraph (lines 448-456).
Q1.4 2) Applying the 0.3 correction coefficient to WAGR syndrome is also problematic
A1.4 We totally agree with these concerns about usage of 0.3 correction coefficient for WAGR syndrome taking into account that this is life-threatening condition which should not be missed. Nevertheless, observed number of patients with WAGR syndrome is still lower than expected. So, our suggestions could be minimal estimations of real situation with actual WAGR syndrome prevalence. Lines 314-320
Q1.5 3) Section 2.5 needs to be removed.
A1.5 Section 2.5 is of the same importance as 2.4. As we do not yet know the most influencing factors for the differences in numbers of collected patients we should start from two different assumptions, the first was that prevalence did not differ, the second in is that it did. We tested 2 hypotheses based on 2 different assumptions, and the second appeared to be less probable because obtained prevalence seemed to be too low in some districts, we inferred, that samples in the district were underrepresented. To make the inferring we should have tested both hypotheses and present both sections, 2.4 and 2.5.
Q1.6 The overview of cases and variants is still useful. I suggest that the paper be modified to note that the number of cases is lower than would be expected based on the published prevalence, suggesting that additional cases are not being referred. Including the epidemiology-based prevalence data (from the second sample) is okay, but more details need to be included about the methods
A1.6 We have modified the name and main accents according to your comments, thank you. However, we earlier already observed cases and variants, but without prediction of what portion of the cases still rests unobserved. And, we also added some details on the methods of epidemiology-based prevalence analysis (lines 115-120)
Q1.7· Ideally, phenotypic information should be included in more detail. The authors note both AN and ‘AN-related’ in the methods, but do not provide details and the frequency of AN vs ‘AN-related’. Ideally, specific diagnosis information should be added the supplemental table and a general breakdown of the numbers in each group should be included in the results. A review of genotype/phenotype correlations/trends would be more valuable.
A1.7 Earlier we performed detailed phenotypic information in a study of genotype-phenotype correlations (PMID: 32467297). The topic of the manuscript is PAX6-associated AN prevalence assessment, so we place the table with phenotypic features in the Materials and Method section as only general and brief sample characteristic.
Introduction
Q1.8· In discussing the mechanisms of PAX6 disruption, deletion of the downstream regulatory region (within ELP4) should be mentioned.
A1.8 Thank you, we added the mention of aniridia due to DRR disruption into the introduction section (line 42) and considerations on the mechanism in the discussion section. Lines 365-368.
Results
Q1.9· · The authors note that they only include PAX6 positive cases. Can any information be provided about the frequency of PAX6 negative cases?
A1.9 Limitations of the study also included PAX6-negative cases. We added this crucial paragraph to the Discussion section. Lines 435-447 By the way now we have ten completed AN diagnostics without defined genetic cause of aniridia. We include into the study only confirmed AN PAX6- associated cases, as we have no way defining what part of the PAX6 negative cases is PAX6-associated ones. So, we could only accept the limitation.
Q1.10· Line 69: ‘Additionally, we uncovered 22 previously unpublished’ is misleading as seems to suggest 22 cases in addition to the 100 newly characterized described in the paragraph below, but I believe the authors mean to say that within this cohort, 22 variants are novel. Suggest merging this with the paragraph below and using something like ‘within this group of intragenic variants, deletion 22 were novel.’
A1.10 Exactly, 22 variants are novel within the cohort of 100 newly characterized probands .line 174-176
Q1.11· Regarding the balanced translocations, discussion of the regulatory region should be included. For example, if the regulatory region is on one side of the breakpoint and PAX6 on the other, this would be a likely mechanism. Alternatively, if the breakpoints fall within the gene, it would result in partial gene deletion. If the breakpoints are not able to be mapped precisely, discussion of the possible mechanisms would be sufficient.
A1.11 Thank you very much, we added the information on the patients with balanced translocations and a discussion on a possible positional effect of the translocations in patients without detected disbalance.lines 193-196
Q1.12· According to the supplemental table, one of the deletions affects the regulatory region only – this should be highlighted.
A1.12 That’s right, thank you, we added a discussion. Lines 365-368.
Q1.13· Section 2.3 and Table 2 could probably be omitted. You could consider a statement that the distribution of variant types did not differ among regions, but it would be unexpected if they did.
A1.13 We totally agree with your opinion and added a sentence about absence of statistically significant differences in distribution of variant types between regions. We added a sentence at the end of precious section lines 234-237
Q1.14 Section 2.4- caution against using ‘existing patients’ to describe predicted prevalence. ‘expected patients’ is more accurate
A1.14 Thank you, we corrected ‘existing’ to ‘expected’.
Q1.15 Section 2.5: I do not feel that these calculations are statistically valid. If comprehensive evaluation was not completed, prevalence cannot be estimated. It is accurate to compare collected cases to predicted based on published prevalence, but the authors cannot use the number of cases collected by referral to calculate prevalence values. Suggest removing this section and focusing on the number of expected missed cases in Section 2.4.
A1.15 We gave the explanation earlier, after general remarks, see above, please.
Q1.16 Instead of comparing variant distribution between regions, it would be more informative to compare the distribution of variant types to other published PAX6 cohorts.
A1.16 Thank you, we did such a comparison earlier and referred to that paper in results and discussion sections lines 380
Q1.17 Similarly, section 2.6 should be modified/removed (see additional notes in methods)
A1.17 In our opinion, this section plays important role since WAGR-syndrome could be clinically undiagnosed in population. We have updated Section 2.5 and Discussion section. (lines 331-337)
Discussion
Q1.18 Lines 234-243 simply restate results and should be removed. The following paragraph of the overall distribution is more relevant.
A1.18 Agreed, removed, thank you.
Materials and Methods:
Q1.19 Per Genes format, this section should be moved between the Introduction and Results, not at the end.
A1.19 Corrected.
Q1.20 Line 324-326 is unclear. The authors state that these 100 patients are both new and part of the initially included samples.
A1.20 Corrected. Now lines 72 and 73
Q1.21 Lines 356-361 seem to belong in results rather than methods
A1.21 We tried to explain the issue earlier, after general comments. As this manuscript topic is prevalence we performed only concise phenotypic characteristics of studied sample and place that in the Methods section.
Q1.22· 4.1.2: second sample. The second paragraph in the section belongs in the results. The first paragraph needs more information on how the prevalence was calculated. Additionally, it is not clear whether the PAX6 cases identified in this group are also included in the group above- this should be specified. I suggest removing the title of ‘second sample’ and replacing with ‘epidemiology study data’ or something similar…
A1.22 We used data previously published by our authors’ collective. Here (now sect. 2.1.2) we briefly described methodology which was used in that publication. In our opinion, this data belongs to the Materials and Methods section rather than to Results which we compare with this previous genetic epidemiology study. And, yes, patients from that epidemiology study belong to our cohort. (lines 125-126)
Q1.23· 4.1.3: first of all, it is not clear why the 0.3 correction was only applied to WAGR and not to AN as a whole. Secondly, I do not believe this is statistically sound- there could be many reasons why patients are not referred for diagnosis, including that less complicated cases are less likely to be referred, which would mean that AN/WAGR cases would be more likely to be referred.
A1.23 We totally agree with you. This section (now 2.1.3) consists only from official statistical information. We moved subsequent information about correction coefficients to the results section with implementation of corrections based on #2 general comments (see above). Concerns about underrepresentation of number of cases was also added to Discussion section. Lines 331-337. Obtained AN/WAGR cases proportion we bring in the line 444.
Q1.24· Table 5 (will need to be renumbered)
A1.24 · Thank you. We have renumbered sections and tables after corrections.
Q1.25 Fovel hypoplasia has a typo- currently states 2244
A1.25 · Thank you. We have corrected the text accordingly.
Q1.26 Would be nice to add glaucoma to this table
A1.26 · Thanks, We added the information on the number of patients with glaucoma.
Q1.27 · This table belongs in results, not methods.
A1.27 · That would be so if we have aimed detailed clinical study, the issue, as we already mentioned, would be a topic of the next manuscript, Here we only characterized a sample of AN patients only because that should characterized.lines 103-105
Comments on the Quality of English Language
The paper is generally well written but has some areas with incorrect word choice/article use. For example:
Q1.28 · Line 36: ‘This condition also cryptically impacts various eye structures in both the anterior and posterior segments’ cryptically is not the correct word choice here- perhaps the authors meant ‘critically’?
A1.28 · Corrected
Q1.29 · Line 41, 42: ‘the WAGR syndrome’ should just be ‘WAGR syndrome’.
A1.29 · Corrected
Q1.30 · Line 398-400: implied is not the correct word choice: ‘followed by Sanger sequencing of the PAX6 gene exons is implied, as described earlier’ and ‘Karyotype analysis are implied in 4 cases
A1.30 · Corrected
Reviewer 2 Report
Comments and Suggestions for Authors
This is an important study addressing poorly understood genetic disease where patients are presented with no iris (Congenital Aniridia: AN) due to a development error caused by the paired box gene (PAX6) mutation. Since this is a rare disease and the prevalence is very low 1:98,943 143 (unpublished author data) and 1:76,335 by Orphanet data, it was a timely research to estimate the prevalence of PAX6-associated AN cases across the entire country, identify potential regional variations, and explore the underlying reasons for such differences. To achieve these goals, the authors have focused on establishing genetically confirmed cases of AN to establish a genetic basis of AN for accurate diagnosis.
The study analyzes PAX6 pathogenic variants and 11p13 chromosome region rearrangements utilizing international database on a cohort of 369 AN patients (285 families, 285 probands) in Russia. In addition, the study identified 100 new families of AN consisting 67 intragenic PAX6 variants and 33 chromosome rearrangement affecting 11p13 region. The study also identified 22 additional small intragenic pathogenic variant of PAX6 gene. Appropriate signed informed consent has been obtained from the study participants and necessary ethical approval has been obtained from appropriate authority to conduct the study. The study took advantage Multiplex Ligation-dependent Probe Amplification (MLPA) analysis of the 11p13 chromosome, PAX6 gene Sanger sequencing, and karyotype analysis.
The study identified a gap related to appropriate genetic screening of cases to identify the actual prevalence of AN in Russian Federation (RF) and 44%-87% underreporting of the AN cases that precludes the possibility of managing the AN patients with necessary medical care.
Overall this is a well designed study. The manuscript reports adequate data analysis and statistical reporting. I have a few minor comments.
1. Explain the abbreviations in its first appearance in the text. In its current form the manuscript appears written for a focus group of ocular geneticist, but the manuscript could be of broader use to other allied scientific communities with focused interest in other rare diseases. Here are a few examples Pax6, MLPA etc.
2. Does the newly characterized 129 AN cases belong to the 369 cases identified by international database search or those are new cases not initially present in the cohort of 369 cases?
3. Page 6, line 216-220: The difference in expected prevalence compared to the global prevalence should be 2.5 fold less (0.2/0.08).
Author Response
This is an important study addressing poorly understood genetic disease where patients are presented with no iris (Congenital Aniridia: AN) due to a development error caused by the paired box gene (PAX6) mutation. Since this is a rare disease and the prevalence is very low 1:98,943 143 (unpublished author data) and 1:76,335 by Orphanet data, it was a timely research to estimate the prevalence of PAX6-associated AN cases across the entire country, identify potential regional variations, and explore the underlying reasons for such differences. To achieve these goals, the authors have focused on establishing genetically confirmed cases of AN to establish a genetic basis of AN for accurate diagnosis.
The study analyzes PAX6 pathogenic variants and 11p13 chromosome region rearrangements utilizing international database on a cohort of 369 AN patients (285 families, 285 probands) in Russia. In addition, the study identified 100 new families of AN consisting 67 intragenic PAX6 variants and 33 chromosome rearrangement affecting 11p13 region. The study also identified 22 additional small intragenic pathogenic variant of PAX6 gene. Appropriate signed informed consent has been obtained from the study participants and necessary ethical approval has been obtained from appropriate authority to conduct the study. The study took advantage Multiplex Ligation-dependent Probe Amplification (MLPA) analysis of the 11p13 chromosome, PAX6 gene Sanger sequencing, and karyotype analysis.
The study identified a gap related to appropriate genetic screening of cases to identify the actual prevalence of AN in Russian Federation (RF) and 44%-87% underreporting of the AN cases that precludes the possibility of managing the AN patients with necessary medical care.
Overall this is a well designed study. The manuscript reports adequate data analysis and statistical reporting. I have a few minor comments.
Q2.1 · Explain the abbreviations in its first appearance in the text. In its current form the manuscript appears written for a focus group of ocular geneticist, but the manuscript could be of broader use to other allied scientific communities with focused interest in other rare diseases. Here are a few examples Pax6, MLPA etc.
A2.1 · Corrected
Q2.2 · Does the newly characterized 129 AN cases belong to the 369 cases identified by international database search or those are new cases not initially present in the cohort of 369 cases?
A2.2 · They are newly included, but belong to the 369 cases, thank you, we added clarification (lines 71-73). Of them, 22 variants are novel (not published in international databases) and 45 previously reported variants (lines 177)
Q2.3 · Page 6, line 216-220: The difference in expected prevalence compared to the global prevalence should be 2.5 fold less (0.2/0.08).
A2.3 · Thank you, corrected – line 341.